# One reporter for in-cell activity profiling of majority of protein kinase oncogenes

Iva Gudernova[1], Silvie Foldynova-Trantirkova[2], Barbora El Ghannamova[2], Bohumil Fafilek[1,3], Miroslav Varecha[1,3], Lukas Balek[4], Eva Hruba[1,3], Lucie Jonatova[4], Iva Jelinkova[1,3], Michaela Kunova Bosakova[1], Lukas Trantirek[2], Jiri Mayer[5], Pavel Krejci[1,3]*

[1]Department of Biology, Faculty of Medicine, Masaryk University, Brno, Czech Republic; [2]Central European Institute of Technology, Masaryk University, Brno, Czech Republic; [3]International Clinical Research Center, St. Anne's University Hospital, Brno, Czech Republic; [4]Department of Experimental Biology, Faculty of Sciences, Masaryk University, Brno, Czech Republic; [5]Department of Internal Medicine, Hematology and Oncology, Masaryk University Hospital, Brno, Czech Republic

**Abstract** In-cell profiling enables the evaluation of receptor tyrosine activity in a complex environment of regulatory networks that affect signal initiation, propagation and feedback. We used FGF-receptor signaling to identify *EGR1* as a locus that strongly responds to the activation of a majority of the recognized protein kinase oncogenes, including 30 receptor tyrosine kinases and 154 of their disease-associated mutants. The *EGR1* promoter was engineered to enhance *trans*-activation capacity and optimized for simple screening assays with luciferase or fluorescent reporters. The efficacy of the developed, fully synthetic reporters was demonstrated by the identification of novel targets for two clinically used tyrosine kinase inhibitors, nilotinib and osimertinib. A universal reporter system for in-cell protein kinase profiling will facilitate repurposing of existing anti-cancer drugs and identification of novel inhibitors in high-throughput screening studies.

*For correspondence: krejcip@med.muni.cz

**Competing interests:** The authors declare that no competing interests exist.

## Introduction

Receptor tyrosine kinases (RTKs) form multiprotein complexes at the cell membrane that mediate signal initiation and propagation, as well as feedback control mechanisms (*Lemmon and Schlessinger, 2010*). While cell-free activity profiling may only uncover chemicals that directly target RTK catalytic function, in-cell profiling confers several additional benefits that could improve the drug development process. First, RTKs are targeted in their natural conformation, with post-translational modifications and in the cell metabolic environment. The protein-protein interactions involved in signal transduction through the RTK-associated signaling complexes, downstream elements, or effector pathways may also be targeted, increasing the chance of success. In-cell profiling may identify biological pathways that naturally oppose the signaling of a certain RTK, and can then be therapeutically exploited (*Wendt et al., 2015*). Furthermore, this approach may also enable the targeting of RTKs in non-signaling states, via interference either with their expression, maturation and transport to the cell membrane or their internalization and degradation. Finally, in-cell activity profiling is applicable to disease-specific in vitro and in vivo models. This is important in the development of therapeutics for chronic diseases caused by pathological RTK signaling, such as diabetes, pulmonary hypertension, chronic kidney disease, or developmental disorders (*Fountas et al., 2015*; *ten Freyhaus et al., 2012*; *Harskamp et al., 2016*; *Laederich and Horton, 2012*), all of which are

poorly represented in current clinic (*Bamborough, 2012*). Lastly, several important oncogenes and downstream targets of RTK signaling, such as RAS, appear not druggable directly (*Cox et al., 2014*) and thus the inhibitors of their signaling may only be discovered via in-cell activity profiling. However, the existing toolkits for in-cell RTK activity profiling provide only partial solutions for the development of RTK inhibitors, as they only focus on a few RTKs or their disease-associated mutants (*Supplementary file 1A*), are technically or instrumentally challenging, or require the development of RTK-specific tools (*Ni et al., 2006*; *Inglés-Prieto et al., 2015*; *Regot et al., 2014*).

A majority of known RTKs activate the RAS/RAF/MEK/ERK MAP kinase signaling module, a pathway that links extracellular mitogenic signals to gene transcription (*Vogelstein et al., 2013*; *Meloche and Pouysségur, 2007*). Hence, the strong effect of ERK on gene transcription could be exploited in the development of reporters that are applicable to the activity profiling of many different RTKs (*Yang et al., 2003*). Here, we report the engineering of one such system, which is based on the promoter sequences of the ERK target gene *EGR1* (*Early growth response 1*). We demonstrate that the *EGR1*-based reporter system is applicable to simple in-cell activity profiling of most protein kinase oncogenes. Additionally, we generate proof-of-concept examples for the use of this system in the identification of novel targets for clinically used protein kinase inhibitors.

## Results and discussion

### Exploitation of the *EGR1* for the activity profiling of fibroblast growth factor receptor (FGFR)

We focused on the particularly strong ERK activation triggered by FGFR signaling in multiple myeloma and rat chondrosarcoma (RCS) cells to identify genes which are upregulated upon ERK activation. The expression profiling of cells treated with FGFR ligand FGF2 identified *Egr1*, *Egr2*, *Nr4A2*, *Dusp6* and *Rgs1* among the most strongly induced genes (*Krejci et al., 2010*; *Buchtova et al., 2015*). The putative promoter sequences of human *EGR1*, *EGR2*, *NR4A2*, *DUSP6* and *RGS1*, located directly upstream of the transcription start sites (*Supplementary file 1B*), were cloned into the promoterless pGL4.17 vector carrying firefly luciferase. In dual-luciferase activity assays performed in RCS cells, *EGR1* promoter showed the strongest response, as it was *trans*-activated approximately 12-fold following FGF2 treatment (*Figure 1A*), and was therefore chosen for subsequent studies. FGF2 induced endogenous EGR1 protein expression in seven different cell types tested, and this phenotype was dependent on ERK activation (*Figure 1A*; *Figure 1—figure supplement 1*).

To identify FGF2-responsive elements, we generated 13 truncated variants of the 2112nt-long human *EGR1* promoter that had originally been cloned into the pGL4.17 vector (−1951/+161 relative to the transcription start site; TSS) (*Figure 1—figure supplement 2*), and subjected these variants to FGF2-mediated *trans*-activation in a dual-luciferase assay (*Figure 1—figure supplement 3*). Four successive rounds of 3- and/or 5-prime sequence shortening and optimization identified a 402nt-long region (−799/–397 relative to TSS) that is critical for FGF2-mediated *trans*-activation (called D-E element) of the *EGR1* promoter (*Figure 1B–D*; *Figure 1—figure supplement 2*). The addition of two extra copies of the D-E element into the *EGR1* promoter (hEGR1-D) enhanced its response to FGF2 by approximately 50% (*Figure 1E*). We named this construct pKrox24(2xD-E_inD)$^{Luc}$ after KROX24, one of the alternative names of EGR1 (*Supplementary file 1C*). The level of pKrox24(2xD-E_inD)$^{Luc}$ FGF2-mediated *trans*-activation in RCS cells peaked at approximately 40 ng/ml FGF2. Treatment of RCS cells with higher doses of FGF2 further elevated ERK activation (*Krejci et al., 2007*), but had a negligible effect on pKrox24(2xD-E_inD)$^{Luc}$ activity (*Figure 1—figure supplement 4*), implying that it is unlikely that further development of the pKrox24(2xD-E_inD)$^{Luc}$ promoter sequence would yield a significant increase in trans-activation capacity. In the five different cell types tested, pKrox24(2xD-E_inD)$^{Luc}$ responded to the activation of FGFR signaling following FGF2 addition, as well as to the chemical inhibition of endogenous FGFR signaling (*Figure 1—figure supplement 5*).

To eliminate inhibitory elements possibly existing within the pKrox24(2xD-E_inD)$^{Luc}$ promoter, a fully synthetic construct was developed, based on *EGR1* promoter and information gained during the pKrox24 development. A human *EGR1* promoter sequence (−1500/+100 bp relative to TSS) was aligned with the corresponding sequences of pig, cow, rat, mouse and chicken *Egr1*, and analyzed by T-Coffee (*Notredame et al., 2000*) to find conserved elements, and by rVista (*Loots and*

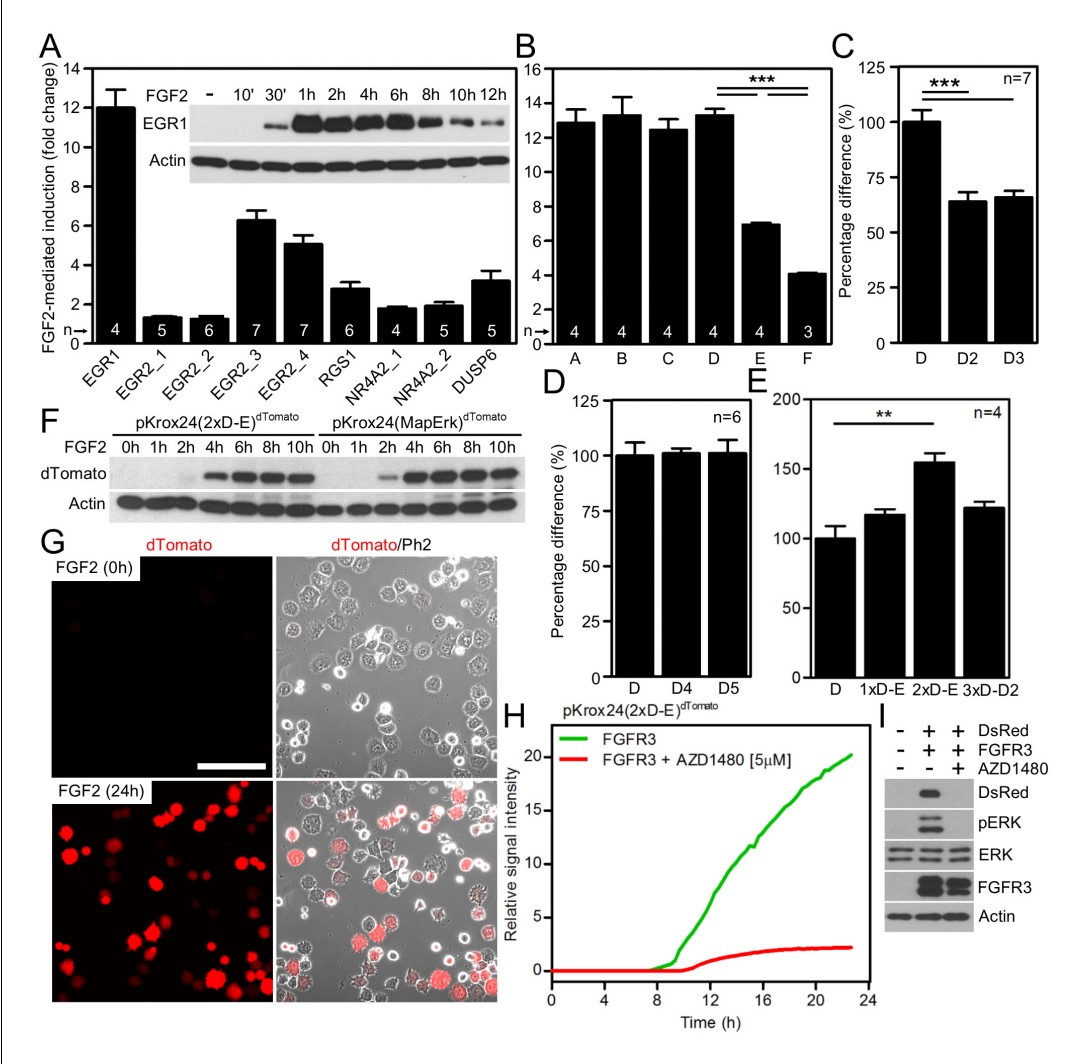

**Figure 1.** Development of luciferase and fluorescent reporters based on a human *EGR1* promoter. (**A**) The activity of various reporters, based on promoters of FGF2-responsive genes, cloned into a pGL4.17 vector expressing firefly luciferase. The FGF2-mediated *trans*-activation (fold-change compared to unstimulated cells) of these reporters in RCS cells was determined by the dual-luciferase assay. Insert, induction of EGR1 protein expression in RCS cells treated with FGF2. (**B–E**) Four consecutive rounds of *EGR1* promoter sequence optimization leading to the pKrox24(2xD-E_inD)[Luc] reporter, including 5'-prime shortening (**B,C**), 3'-prime shortening (**D**), and addition of repetitive D-elements (**E**) to the originally cloned *EGR1* promoter (vectors outlined in *Figure 1—figure supplement 2*). The presented data were generated through dual-luciferase assays in RCS cells, with 'n' describing the number of independent experiments. Statistically significant differences are highlighted (Student's t-test; **p<0.01, ***p<0.001). (**F, G**) FGF2-mediated induction of dTomato protein expression (**F**) and fluorescence (**G**) in RCS cells transiently transfected with pKrox24(2xD-E)[dTomato] or pKrox24(MapErk)[dTomato] reporters. Bar, 150 μm. (**H**) Transactivation of pKrox24(2xD-E)[dTomato] in RCS cells induced by forced expression of the constitutively active FGFR3 K650M mutant, determined by live cell imaging of dTomato fluorescence over 24 hr. The dTomato induction was suppressed by the FGFR inhibitor AZD1480. (**I**) Immunoblot validation of DsRed induction and ERK phosphorylation (p) in RCS cells transfected with FGFR3 K650M mutant together with pKrox24(2xD-E)[DsRed] for 16 hr. Actin and total ERK levels served as loading controls.

The following figure supplements are available for figure 1:

**Figure supplement 1.** FGF2 induces EGR1 expression dependent on ERK MAP kinase.

**Figure supplement 2.** Schematic outline of human *EGR1* promoter sequences cloned into the promoterless pGL4.17 vector carrying firefly luciferase, and analyzed for FGF2-mediated *trans*-activation as shown by *Figure 1A–E*.

**Figure supplement 3.** Analysis workflow of the dual-luciferase assay.

*Figure 1 continued on next page*

*Figure 1 continued*

**Figure supplement 4.** The extent of pKrox24(2xD-E_inD)^Luc reporter *trans*-activation with increasing FGF2 concentrations in RCS cells.

**Figure supplement 5.** Validation of pKrox24(2xD-E_inD)^Luc reporter in cellular models to FGFR signaling.

**Figure supplement 6.** Generation of pKrox24(MapErk) reporters.

**Figure supplement 7.** Comparison of transactivation capacity and basal activity of pKrox24(MapErk)and pKrox24(2xD-E_inD) reporters.

**Figure supplement 8.** FGF-mediated transactivation of constructs containing D-E or MapErk promoter elements combined with dTomato or DsRed reporters.

*Ovcharenko, 2004*) to identify transcription factor binding sites. The identified elements, together with previously discovered sequences (*Wang et al., 2010*), were mapped onto the *EGR1* promoter and cloned into a pGL4.26 vector containing a minimal promoter and firefly luciferase. The resulting reporter was named pKrox24(MapErk)^Luc (*Figure 1—figure supplement 6*). When compared to pKrox24(2xD-E_inD)^Luc, pKrox24(MapErk)^Luc showed no additional increase in FGF2-mediated *trans*-activation, but possessed a significantly lower basal activity in RCS and 293T cells (*Figure 1—figure supplement 7*).

Additionally, constructs containing five copies of MapErk or two copies of D-E elements, serving as minimal promoters, were generated with either dTomato or DsRed reporter (*Supplementary file 1C*). These reporters responded well to FGF2-mediated *trans*-activation following transfection into RCS cells, shown by immunoblot analyses of dTomato and DsRed expression, and live cell imaging (*Figure 1F,G*; *Figure 1—figure supplement 8*). Expression of the constitutively active FGFR3 mutant K650M (*Naski et al., 1996*) in RCS cells induced dTomato or DsRed expression, and this expression was reversed by treatment with AZD1480, a chemical inhibitor of FGFR3 (*Scuto et al., 2011*) (*Figure 1H,I*; *Figure 1—figure supplement 8C*).

## Protein kinases induce EGR1 protein expression in cells

Next, we investigated whether RTKs other than FGFRs induce EGR1 expression in cells. A total of 37 full-length human wild-type (wt) RTKs were cloned into the pcDNA3.1 vector in frame with a C-terminal V5/6xHis epitope, expressed in 293T cells, and verified by immunoblot (*Figures 2* and *3*). Site-directed mutagenesis was used to generate the major mutants of each RTK associated with human disease, obtained via surveys of published literature, or selected from the catalogue of mutations associated with human cancers or inherited conditions available from the Sanger Cosmic (*Forbes et al., 2015*) and OMIM databases. As RTKs auto-phosphorylate upon activation (*Bae and Schlessinger, 2010*), a phosphorylation-specific RTK antibodies were used to estimate the spontaneous or ligand-induced activation of expressed RTKs (*Figures 2* and *3*; *Supplementary file 1D*). A total of 254 wt and mutant RTK variants were prepared this way, expressed in 293T cells, and analyzed for EGR1 induction. The results showed that 30 wt RTKs (81%) and 154 (71%) of their mutants induced EGR1 when expressed in 293T cells (*Figure 4*).

Three wt RTKs (FGFR3, TIE, VEGFR1) and 27 mutants induced EGR1 expression but were not found to be phosphorylated (*Figures 2–4*). This is likely due to the fact that in active RTKs some phosphotyrosines are differentially phosphorylated and thus may not be identified by antibodies designed for specific motifs. The most notable example is FGFR3, as all five active mutants associated with skeletal dysplasia and cancer (*Kant et al., 2015*; *Passos-Bueno et al., 1999*; *Carter et al., 2015*) induced EGR1, but only K650M FGFR3 was found to be phosphorylated by an antibody recognizing FGFR phosphorylation at Y653/Y654. The pseudokinases lacking catalytic activity (ROR1, ROR2, RYK, ERBB3) were among the RTKs that did not induce EGR1, along with RTKs that did not autophosphorylate after expression in 293T cells (TYRO3, INSRR, ROS1) (*Figures 2* and *3*). Furthermore, 36 mutants failed to induce EGR1 despite being derived from RTKs that induce EGR1 expression. The majority of these mutants (83%) were kinase-inactive mutants (*Figures 2–4*). The remaining six mutants did not induce EGR1 because of weak activation (DDR1^R896Q, ERBB4^E836K, RET^E768N) or due to an unknown reason (RON^R470C, RON^R1231C, PDGFRB^D850N). Overall, the RTK activation

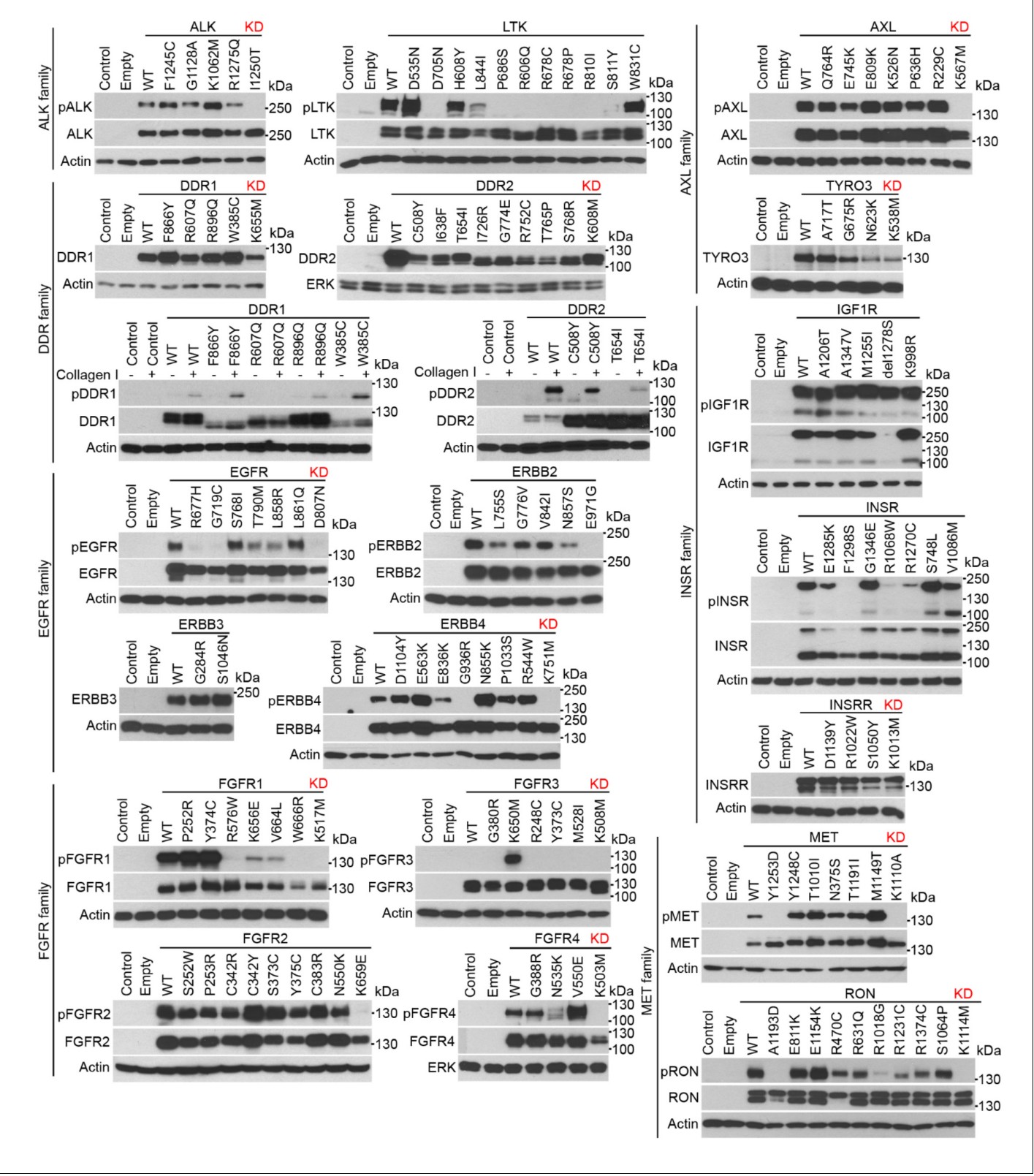

**Figure 2.** RTK cloning and validation (part 1). Full-length human RTK cDNA was cloned into pcDNA3.1 vectors and equipped with a C-terminal V5/His epitope. Mutants were created by site-directed mutagenesis. The RTKs were expressed in 293T cells, and their activation was probed by immunoblot with antibodies that recognize the given RTK only when it is phosphorylated (p) at a specific motif, with exception of phosphorylated DDR1 and DDR2 which were detected with pan-pY antibody. A total of 37 wild-type (WT) RTKs and 241 of their mutants were obtained, including disease-associated

*Figure 2 continued on next page*

*Figure 2 continued*
loss-of-function and gain-of-function mutants, and experimental kinase-inactive mutants (KD). Treatment with the cognate ligands of DDR1, DDR2, KIT, and VEGFR2 was used for the activation of these RTKs.

correlated with EGR1 induction in 96% (154 out of 160) of the tested wt and mutant RTKs. Thirteen additional non-receptor tyrosine kinases, serine/threonine kinases C-RAF and B-RAF as well as RAS small GTPase, were subjected to the same analyses (*Figure 4—figure supplement 1*). Taken together, we have demonstrated that, apart from the JAK and MAPKK kinases not evaluated in this study, all of the protein kinase oncogenes recognized to date (*Vogelstein et al., 2013*) are capable of inducing EGR1 expression in 293T cells.

## pKrox24 reporters can be used to identify novel targets for clinically used kinase inhibitors

One application of pKrox24 reporters is the identification of novel targets for clinically used kinase inhibitors, which could help repurpose existing anti-cancer drugs or uncover the molecular mechanisms underlying the side-effects they cause in patients. Chronic myeloid leukemia (CML) is a clonal myeloproliferative disorder characterized by a t(9;22)(q34;q11) translocation that produces a cytoplasmic BCR-ABL fusion protein with constitutive tyrosine kinase activity (*Zhao et al., 2002*). The suppression of BCR-ABL catalytic activity with tyrosine kinase inhibitors (TKI) has greatly improved CML prognosis, effectively turning a once fatal cancer into a manageable chronic disease. Several generations of BCR-ABL TKIs have been developed to improve efficacy and overcome the BCR-ABL resistance to first generation TKIs caused by mutations and gene amplifications (*Hochhaus et al., 2008*; *Cortes et al., 2013*). However, some TKIs, such as ponatinib, can cause severe toxicity in CML patients and even lead to discontinuation of the therapy (*Modugno, 2014*). The reasons why these BCR-ABL TKI side-effects occur are not clear, and in this way, the elucidation of how TKIs affect physiological tyrosine kinase signaling is of major interest to CML research. To identify novel targets, we evaluated the activity of five clinically used BCR-ABL TKIs, that is ponatinib, imatinib, dasatinib, bosutinib and nilotinib, against a panel of 28 wt RTKs. Different TKI concentrations were used to assess the inhibition of BCR-ABL activity as well as cell toxicity (*Figure 5—figure supplement 1A*). *Figure 5A* shows that all tested TKIs inhibited RTKs that had already been reported as targets in literature (*Supplementary file 1E*), with exception of LTK and INSR, which were identified as two novel targets for nilotinib (*Figure 5A*; *Figure 5—figure supplement 1E*).

Osimertinib (AZD9291) is recently described inhibitor of EGFR catalytic activity, and was approved for clinical use in lung carcinoma in 2015 (*Cross et al., 2014*; *Greig and Approval, 2016*). Osimertinib is a mutant-selective EGFR inhibitor, with 200-fold selectivity for EGFR mutants T790M and L858R over the wt EGFR (*Cross et al., 2014*; *Finlay et al., 2014*; *Jiang and Zhou, 2014*). Crystallographic studies indicate that osimertinib binds to the outer edge of the EGFR ATP binding pocket through a covalent bond with Cys797 (*Yosaatmadja et al., 2015*). Although these data provide no clear explanation for EGFR mutant versus the wt selectivity (*Cross et al., 2014*; *Finlay et al., 2014*; *Jiang and Zhou, 2014*; *Yosaatmadja et al., 2015*), osimertinib is expected to possess a very narrow spectrum of RTK specificity, limited to EGFR and the closely related ERBB2 and ERBB4. We tested this prediction by evaluating osimertinib activity against 30 wt RTKs and 116 of their active mutants with a pKrox24(2xD-E_inD)$^{Luc}$ luciferase assay in 293T cells. We observed inhibitory activity for osimertinib against EGFR, ERBB2 and ERBB4, but not for the other 26 RTKs and 99 mutants (*Figure 5B*). However, LTK was an exception, as it appeared to be inhibited by osimertinib in both wt and mutant forms. These included the D535N and L844I mutants, which associate with multiple myeloma and stomach carcinoma (*Hucthagowder et al., 2012*; *Kubo et al., 2009*), respectively, and the W831C and H608Y substitutions found in the Cosmic and VarSome databases. The osimertinib activity was confirmed by suppression of the autophosphorylation of LTK expressed in 293T cells, and by inhibition of LTK-mediated phosphorylation of recombinant STAT1 substrate in cell-free kinase assays (*Figure 5C*).

The presented pKrox24 technology enables rapid in-cell profiling of a majority of the known protein kinase oncogenes via simple and versatile reporters based on the activity of a downstream protein kinase signaling target. While the luciferase reporters may be applied to tractable cell models

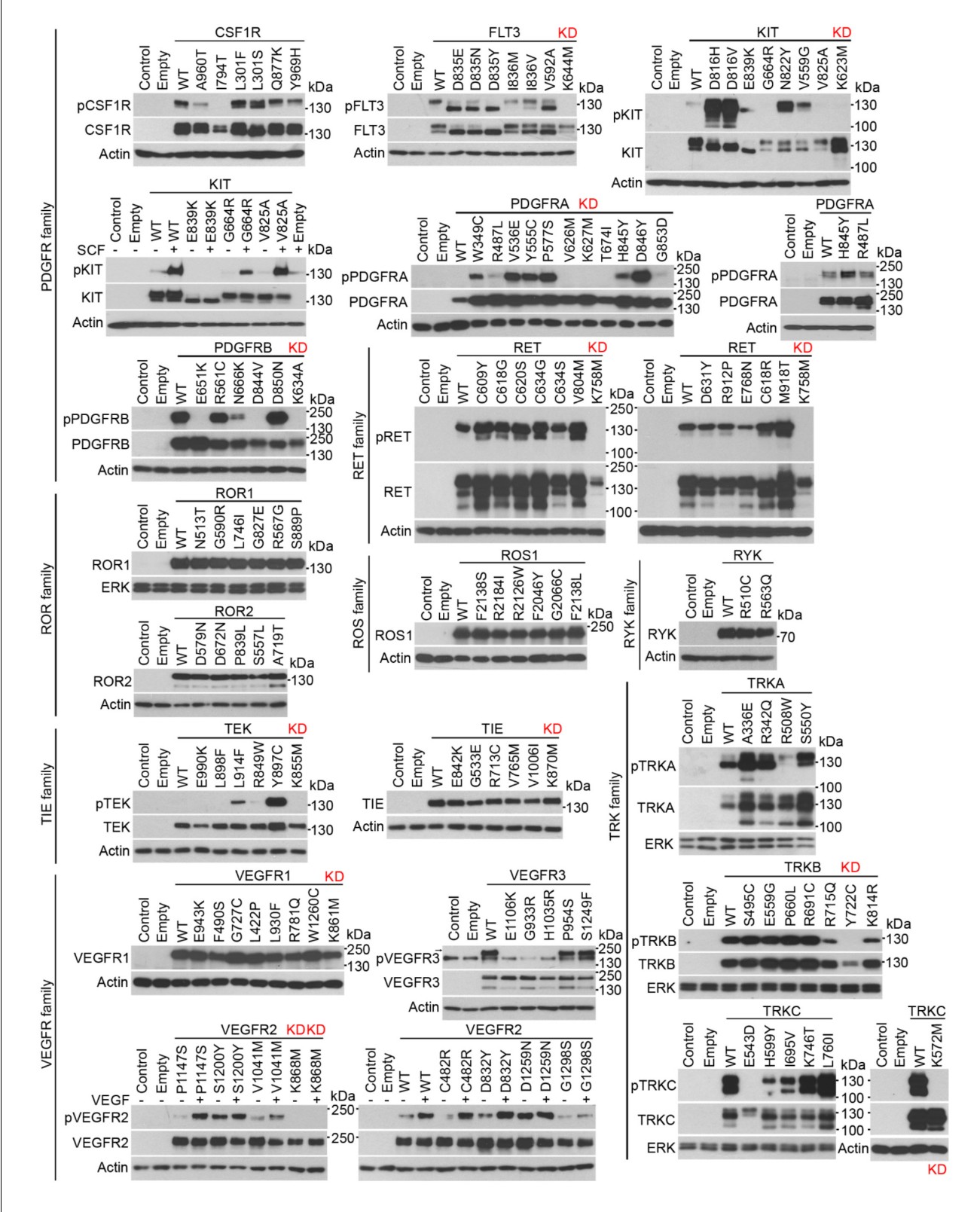

**Figure 3.** RTK cloning and validation (part 2).

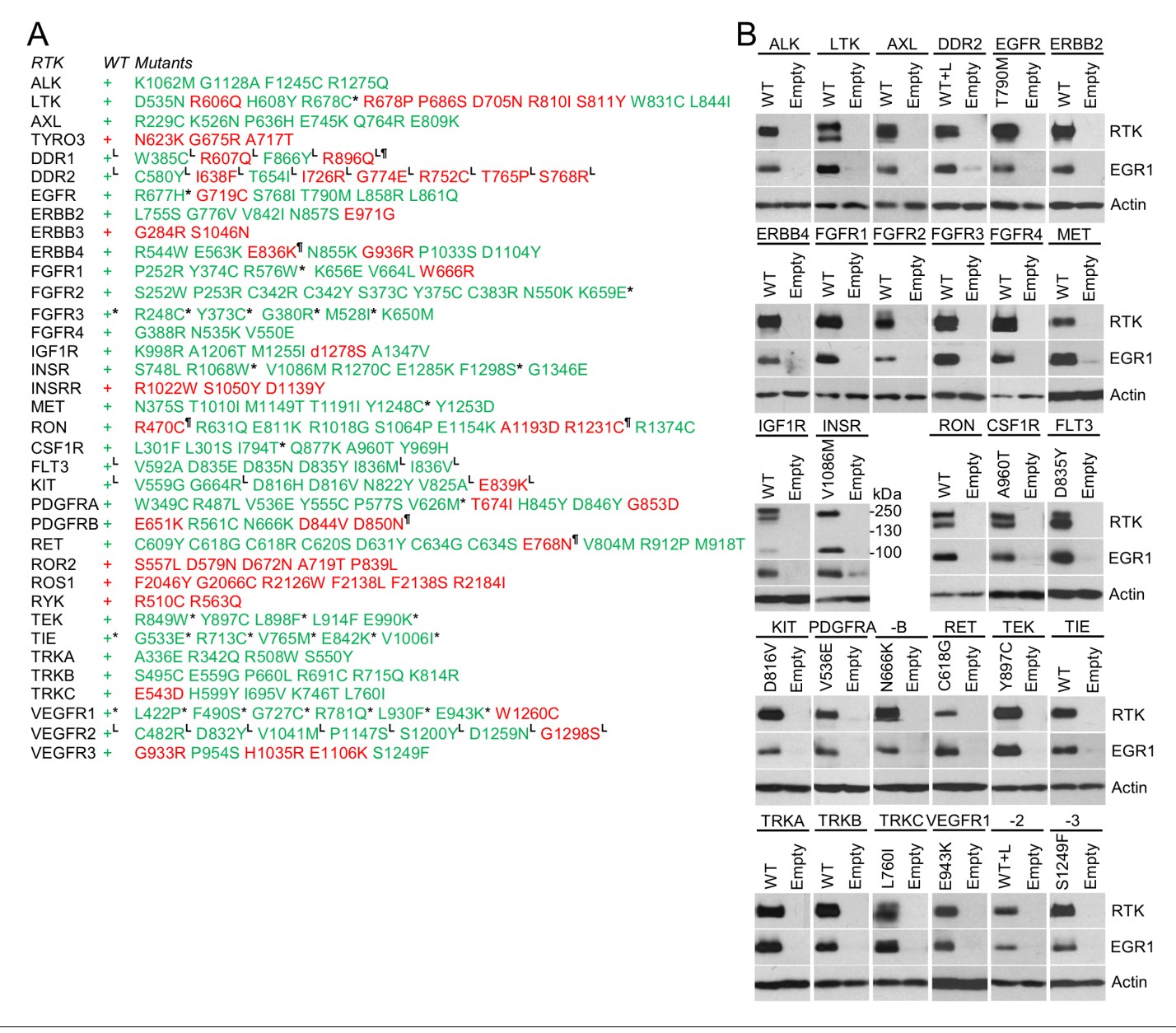

**Figure 4.** RTKs induce EGR1 protein expression. (**A**, **B**) Immunoblot analyses of EGR1 induction in 293T cells transfected with wild-type (WT) or mutated RTKs for 24 hr. Cells transfected with empty plasmids serve as the transfection control, and actin serves as the loading control. (**A**) Green, RTK induces EGR1; red, no EGR1 induction by the RTK; * RTKs that induced EGR1 but were not autophosphorylated (**Figures 2** and **3**); ¶ RTKs that were autophosphorylated but did not induce EGR1; L RTKs activated by the addition of their cognate ligands.

The following figure supplement is available for figure 4:

**Figure supplement 1.** EGR1 expression induced by non-receptor tyrosine kinases, serine/threonine kinases C-RAF and B-RAF, and RAS small GTPase.

to repurpose existing protein kinase inhibitors, the application of fluorescent pKrox24 reporters to high-throughput screening (HTS) of compound libraries offers a major advantage. Cells can be viewed any time during the screening, and this characteristic of HTS would enable researchers to detect false-positive hits based on the inhibition of dTomato and DsRed expression by cell-toxic compounds through mechanisms unrelated to the target protein kinase. Hence, these reporters could improve the interpretation of HTS screening data by readily eliminating false-positive hits.

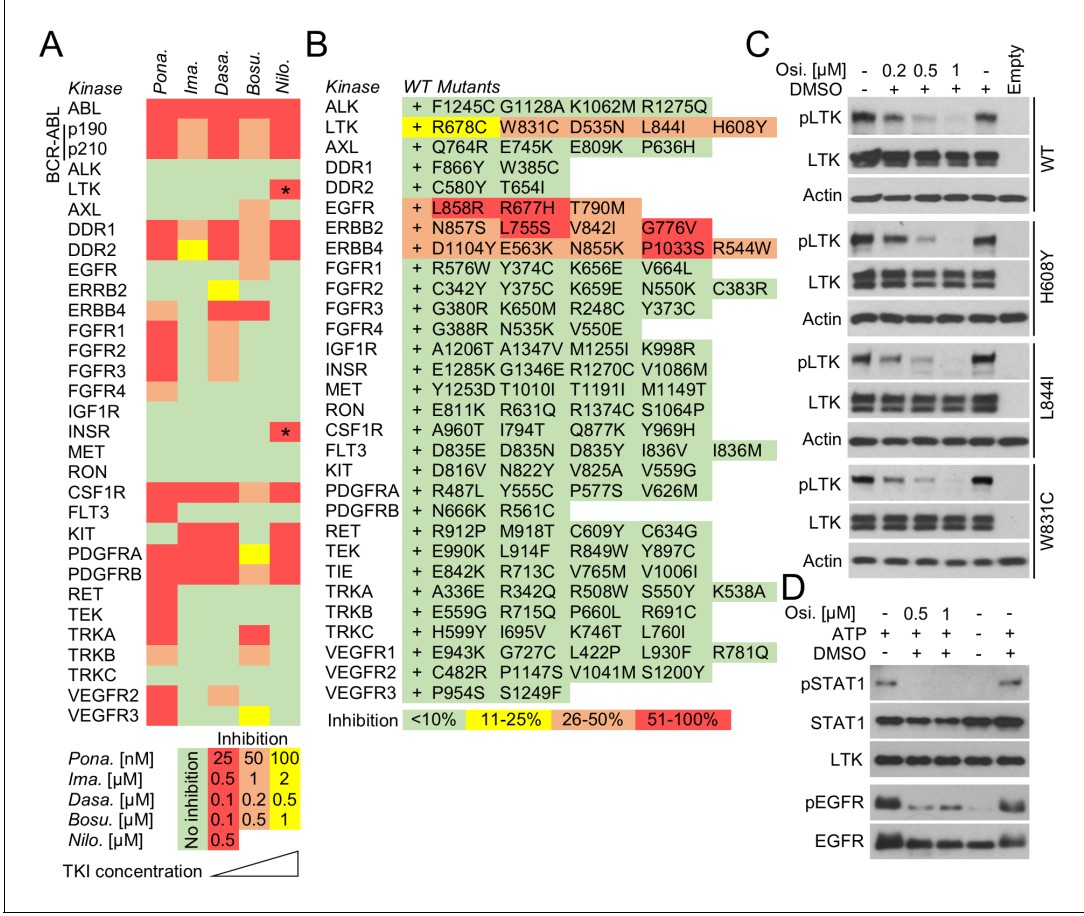

**Figure 5.** In-cell RTK activity profiling with BCR-ABL and EGFR inhibitors. (**A**) Activity of BCR-ABL inhibitors ponatinib (*Pona.*), imatinib (*Ima.*), dasatinib (*Dasa.*), bosutinib (*Bosu.*), and nilotinib (*Nilo.*) against 28 wild-type RTKs, evaluated in 293T cells transfected with RTKs and treated with inhibitors for 20–24 hr. The panel compiles data from immunoblot detections of activated RTKs, each treated with inhibitor concentrations derived from the experiments shown in *Figure 5—figure supplement 1*. Only one concentration is shown for nilotinib due to its cell toxicity at higher concentrations. Asterisks highlight the previously unreported nilotinib targets LTK and INSR (*Supplementary file 1E*; *Figure 5—figure supplement 1*). (**B**) Activity profiling of 30 wild-type (wt) RTKs and 116 of their active mutants in the presence of 0.5 µM osimertinib. 293T cells were transfected with RTK vectors together with pKrox24(2xD-E_inD)^Luc 24 hr before osimertinib treatment (for 24 hr). The colors reflect the osimertinib-mediated inhibition of pKrox24(2xD-E_inD)^Luc *trans*-activation induced by a given RTK, relative to cells untreated with osimertinib. Basal levels of osimetrinib-mediated inhibition of pKrox24 (2xD-E_inD)^Luc were obtained from cells transfected with empty plasmid and then subtracted from the data. (**C**) 293T cells were transfected with wt LTK or its mutants, and treated with osimertinib (Osi.) for 24 hr. The LTK autophosphorylation (p) reflect LTK activity. Total LTK and actin serve as loading controls. (**D**) Cell-free kinase assays were carried out with recombinant LTK or EGFR and osimertinib added to the kinase reaction. Phosphorylation (p) of a recombinant STAT1 and autophosphorylation was used to detect LTK and EGFR activation, respectively. Samples with omitted ATP serve as negative controls for kinase activity.

The following figure supplement is available for figure 5:

**Figure supplement 1.** Analyses of cytotoxicity and kinase activities of BCR-ABL and EGFR inhibitors.

## Methods

### Cell culture, transfection and luciferase reporter assay

NIH3T3 cells (RRID:CVCL_0594) and 293 T cells (RRID:CVCL_0063) were obtained from ATCC (Manassas, VA). hiPSC cell line AM13 was generated as described before (*Krutá et al., 2014*). hESC (CCTL14; RRID:CVCL_C860) cells were prepared as described before (*Dvorak et al., 2005*). RCS cells (RRID:CVCL_S122), KMS11 (RRID:CVCL_2989) and LP1 (RRID:CVCL_0012) cells were obtained as described before (*Krejci et al., 2010*). All used cell lines were routinely evaluated for mycoplasma

contamination using DAPI staining and confocal microscopy, and were mycoplasma free. Cells were propagated in DMEM media, supplemented with 10% FBS and antibiotics (Invitrogen, Carlsbad, CA). hESC and hiPSC cells were propagated in feeder-free conditions. For 293T growth assays, 3 × 10 cells were grown in 24-well tissue culture plates for 1 day, and the cells were treated with inhibitors. After 24 hr the cell numbers were determined by cell counter (Beckman Coulter, Brea, CA). Chemicals were obtained from the following manufacturers: FGF2, SCF, FLT3 ligand, VEGF (RnD Systems, Minneapolis, MN); collagen type 1 (Santa Cruz Biotechnology, Santa Cruz, CA); PD0325901 (Tocris Bioscience, Bristol, UK); ponatinib, imatinib, dasatinib, bosutinib, nilotinib, osimertinib, AZD1480, BGJ398 (Selleckchem, Houston, TX); heparin (Sigma-Aldrich, St. Louis, MO). Cells were transfected either by using FuGENE6 transfection reagent (Roche, Basel, Switzerland), polyethylenimine (Sigma-Aldrich) or electroporation with the Neon Transfection System (Invitrogen). For the luciferase reporter assay, cells were also transfected with a vector expressing firefly luciferase and a vector expressing *Renilla* luciferase under the control of a constant promoter (pRL-TK) at a 3/1 ratio. Luciferase signal was quantified 20–24 hr later using the Luciferase or Dual-Luciferase Reporter Assay (Promega, Madison, WI). Osimertinib screening in 293T cells was carried out with pKrox24(2xD-E)$^{Luc}$only, which was transfected at a 1/3 ratio together with the RTK-expressing vector. The cells were treated with 0.5 µM osimertinib 24 hr after transfection, and luciferase signal was determined 24 hr later.

## Plasmid cloning and mutagenesis

Vectors (pcDNA3.1) carrying C-terminally V5-tagged RTKs were generated by cloning full-length human RTK cDNA into a pcDNA3.1/V5-His TOPO TA vector (Invitrogen). Site-directed mutagenesis was carried-out according to the manufacturer's protocol (Agilent, Santa Clara, CA). Vectors (pCR3.1) carrying N-terminally FLAG-tagged p190 and p210 variants of BCR-ABL were generated by cloning full-length human BCR-ABL p190 cDNA (source pSG5-P190) and full-length human BCR-ABL p210 cDNA (source Bcr/Abl P210LEF) into a pCR3.1 vector (Invitrogen) containing a PGNQNMDYKDDDDK amino acid coding sequence between BamHI and EcoRI in the multiple cloning site. The source vectors pSG5-P190 (Addgene, Cambridge, MA; plasmid #31285) and Bcr/Abl P210LEF (plasmid #38158) were a gift from Nora Heisterkamp (*Yi et al., 2008*; *Kweon et al., 2008*). Promoter regions of *EGR1*, *EGR2*, *RGS1*, *NR4A2* and *DUSP6* were amplified from hESC genomic DNA by PCR and ligated to a pGL4.17 vector (Promega). All EGR1 promoter fragments of different lengths (hEGR1-B - hEGR1-F) were amplified from hEGR1-A by PCR and inserted into vector pGL4.17. The construct pKrox24(1xD-E_inD)$^{Luc}$ was prepared by inserting the active element D-E, amplified from hEGR1-A by PCR, into the hEGR1-D construct. pKrox24(2xD-E_inD)$^{Luc}$ was obtained by cloning synthetic DNA corresponding to two copies of the active element D-E to KpnI site of the hEGR1-D construct. pKrox24(3xD-D2_inD)$^{Luc}$ was obtained by cloning three copies of the active element D-D2 as a synthetic gene by KpnI into the hEGR1-D construct. pKrox24(MapErk)$^{Luc}$ was prepared by inserting synthetic DNA corresponding to five copies of a designed MapErk sequence (listed in *Figure 1—figure supplement 6*) into KpnI, HindIII sites of the pGL4.26 vector (Promega). Two copies of synthetic DNA corresponding to element D-E were cloned into HindIII, SalI sites of pDsRed-Express-DrVector (Clontech) to obtain pKrox24(2xD-E)$^{DsRed}$. pKrox24 (MapErk)$^{DsRed}$ was prepared by cloning of five copies of designed MapErk sequence into EcoRI, BamHI sites of the pDsRed-Express-Dr vector. pKrox24(2xD-E)$^{dTomato}$ and pKrox24(MapErk)$^{dTomato}$ were generated by swapping of dTomato cDNA from ptdTomato Vector (Clontech) to pCLuc-Basic2 by HindIII,NotI and BamHI,NotI sites, respectively. 2xD-E element was cloned to pCLuc-Basic2 by EcoRI,EcoRV sites, MapErk element was cloned into EcoRI,XhoI sites. *Supplementary file 1C* lists all expression vectors used in the study; *Supplementary file 1F* lists all PCR primers with marked restriction sites used for plasmid generation.

## Immunoblotting

Cells were harvested into the sample buffer (125 mM Tris-HCl pH 6.8, 20% glycerol, 4% SDS, 5% *β*-mercaptoethanol, 0.02% bromophenol blue). Samples were resolved by SDS-PAGE, transferred onto a PVDF membrane and visualized by chemiluminiscence (Thermo Scientific, Rockford, IL). *Supplementary file 1D* lists the antibodies used in the study. Kinase assays were performed with 200 ng of recombinant EGFR or LTK (SignalChem, Richmond, CA) in 50 µl of kinase buffer (60 mM

HEPES pH 7.5, 3 mM MgCl$_2$, 3 mM MnCl$_2$, 3 µM Na$_3$VO$_4$, 1.2 mM DTT) in the presence of 10 µM ATP for 60 min at 30°C. Recombinant STAT1 was from Cell Science (Newbury port, MA).

## Live cell imaging

Time-lapse microscopy experiments with living cells were conducted using either an automated incubation microscope BioStation CT (Nikon, Tokio, Japan) or a confocal laser-scanning microscope Carl Zeiss LSM 700 (Carl Zeiss, Jena, Germany) equipped with an atmospheric chamber. Phase contrast and fluorescence signal images were automatically acquired every 15 min during a 24 hr time period. Images were then processed and analyzed in either Nikon BioStation CT or Carl Zeiss ZEN 2 software. Phase contrast and fluorescence images were exported into Microsoft Publisher for the preparation of publication figures.

## Acknowledgements

The authors wish to thank Iva Vesela, Iveta Cervenkova, Arelys Puerta and Jorge Martin for their assistance with plasmid cloning, and Miriam Minarikova, Zaneta Konecna and Pavel Nemec for their excellent technical assistance. This work was supported by Ministry of Education, Youth and Sports of the Czech Republic (KONTAKT II LH15231, CZ.1.05/3.1.00/14.0324); Technology Agency of the Czech Republic (TG02010048); Grant Agency of Masaryk University (0071–2013); Czech Science Foundation (GA17–09525S); Ministry of Health of the Czech Republic (15-33232A, 15-34405A); National Program of Sustainability II (MEYS CR: LQ1605 and LQ1601) and European Union ICRC-ERA-HumanBridge (No. 316345); and by funds from the Faculty of Medicine at Masaryk University to junior researcher MKB. SFT received support from the SoMoPro II Programme G4 target, which was co-financed by the European Union and the South-Moravian Region (Note: This publication reflects only the author's views and the Union is not liable for any use that may be made of the information contained therein). LT was supported by the career development grant from the European Organization for Molecular Biology (IG2535) and by the Marie-Curie Re-integration grant (ECOPOD). IG was supported by Specific University Research Grant at Masaryk University (MUNI/A/0810/2016; Ministry of Education, Youth and Sports of the Czech Republic).

## Additional information

### Funding

| Funder | Grant reference number | Author |
|---|---|---|
| Ministry of Education, Youth and Sports of the Czech Republic | MUNI/A/0810/2016 | Iva Gudernova |
| SoMoPro II Programme | G4 target | Silvie Foldynova-Trantirkova |
| Faculty of Medicine Masaryk University | Junior grant | Michaela Kunova Bosakova |
| European Molecular Biology Organization | IG2535 | Lukas Trantirek |
| Marie-Curie Re-integration grant | ECOPOD | Lukas Trantirek |
| Ministry of Education, Youth and Sports of the Czech Republic | KONTAKT II LH15231 | Pavel Krejci |
| Grant Agency of Masaryk University | 0071-2013 | Pavel Krejci |
| Ministry of Health of the Czech Republic | 15-33232A | Pavel Krejci |
| European Union ICRC-ERA-Human Bridge | 316345 | Pavel Krejci |
| Grant Agency of the Czech Republic | GA17-09525S | Pavel Krejci |

| Technology Agency of the Czech Republic | TG02010048 | Pavel Krejci |
| Ministry of Health of the Czech Republic | 15-34405A | Pavel Krejci |
| Ministry of Education, Youth and Sports of the Czech Republic | CZ.1.05/3.1.00/14.0324 | Pavel Krejci |

The funders had no role in study design, data collection and interpretation, or the decision to submit the work for publication.

## Author contributions

IG, BF, Conceptualization, Data curation, Formal analysis, Writing—original draft, Writing—review and editing; SF-T, Conceptualization, Data curation, Formal analysis, Supervision, Writing—original draft, Writing—review and editing; BEG, LB, EH, LJ, IJ, MKB, Data curation; MV, Conceptualization, Data curation, Visualization; LT, Conceptualization, Resources, Supervision, Funding acquisition, Methodology, Writing—original draft, Writing—review and editing; JM, Conceptualization, Resources, Funding acquisition, Writing—original draft, Writing—review and editing; PK, Conceptualization, Data curation, Formal analysis, Supervision, Funding acquisition, Writing—original draft, Writing—review and editing

## Author ORCIDs

Pavel Krejci, http://orcid.org/0000-0003-0618-9134

## Additional files

### Supplementary files

• Supplementary file 1. Supplementary tables containing (A) Commercial providers of RTK activity profiling; (B) Nucleotide sequences cloned into the promoterless pGL4.17 vector expressing firefly luciferase; (C) Expression vectors used in the study; (D) Antibodies used in the study; (E) Literature survey of anti-RTK activity of BCR-ABL TKIs; (F) Primers used for reporter construction.

• Supplementary file 2. Supplementary file contains numerical data for *Figure 1A,B,C,D and E*; *Figure 1—figure supplements 3*, *4*, *5A, B*, *7A and B*; and *Figure 5—figure supplement 1*.

• Supplementary file 3. Supplementary file contains numerical data for *Figure 5B*.

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
