## [Decision Letter]

Thank you for submitting your article "One reporter for in-cell activity profiling of majority of protein kinase oncogenes" for consideration by *eLife*. Your article has been favorably evaluated by Tony Hunter as the Senior Editor and three reviewers, one of whom is a member of our Board of Reviewing Editors. The reviewers have opted to remain anonymous.

The reviewers have discussed the reviews with one another and the Reviewing Editor has drafted this decision to help you prepare a revised submission.

Summary:

The authors describe an optimized EGR1-based reporter that can be used to measure the activity of many receptor tyrosine kinases. The strength of the system is that activity can be used in living cells and that it is amenable to high throughput screening. It is therefore likely that the reporter will be useful for many investigators.

Essential revisions:

1) Can the output of different kinases be quantitatively compared using this EGR1-driven reporter?

2) To what extent do different kinases turn on EGR1? Is it possible to normalize for extent of expression and specific activity of the overexpressed kinase?

3) FGFR inhibitors only partially block reporter activity caused by FGFR. Is this because of autocrine growth factor production and activation of ERK by another pathway? How does this limit the usefulness of the reporter?

4) The use of over expression in 293T cells to provide ligand-independent signaling is potentially a problem because the constitutive activity of these ligand-regulated receptors may not reflect their physiological or pathophysiological activity? Thus, while the reporter assay may be useful for identifying inhibitors, one wonders if it may be limited in other regards. This issue deserves some comment.

5) The authors point to the uncertainty surrounding the clinical toxicity of the BCR-ABL inhibitor ponatinib (subsection “pKrox24 reporters can be used to identify novel targets for clinically used kinase inhibitors”, first paragraph) as a potential use for their assay, but do not indicate whether their analysis of ponatinib reveals new targets that might explain the side effects arising from its ability to target other RTKs.

6) While the authors identify new targets for nilotinib (LTK and INSR), this discovery was made without the EGR1-based reporter. Inclusion of data for an application that utilized the EGR1-based reporter would be an effective way to demonstrate the utility of this system.

---

## [Author Response]

Essential revisions:

*1) Can the output of different kinases be quantitatively compared using this EGR1-driven reporter?*

In Figure 1—figure supplement 4, we used increasing concentrations of FGF-receptor (FGFR) ligand FGF2 to activate endogenous FGFR signaling in cells, in order to test to what extent the level of Krox24 transactivation correlates with differing levels of activation of endogenous FGFR signaling. We demonstrate that Krox24 transactivation gradually increases with 1-40 ng/ml FGF2, but peaks and plateaus at 40-100 ng/ml of FGF2. This suggests that the Krox24 reporters may have a limited capacity to quantitatively record the differences in RTK activation, especially in RTKs which are strong activators of ERK pathway, such as the FGFRs. In the manuscript revision, we tested the ability of Krox24 reporters to record differences in RTK activation by two experiments: (A) by determining whether different mutants belonging to one RTK transactivate Krox24 differently, corresponding to their known activity; (B) by determining whether the two groups of unrelated wt RTKs, belonging to strong or weak ERK activators, transactivate Krox24 reporters differently. The data presented by Figure 6 demonstrate that, in both cases, the relative differences in RTK activity can be distinguished by the levels of EGR1 expression and Krox24 reporter transactivation.

Author response image 1.(**A**) 293T cells were transfected with pKrox24 reporters together with empty vector and vectors expressing wild-type (WT) FGFR3 or its activating mutants G380R (associated with achondroplasia) and Y373C (associated with thanatophoric dysplasia). Note the differences in pKrox24 transactivation (upper graphs), and EGR1 induction (lower blot), which correspond to well documented differences in relative activity of the FGFR3 variants (see, for instance Krejci et al., PLoS *ONE*. 2008; 3: e3961), which is as follows: wt FGFR3<G380R<Y373C. K508M; kinase-dead FGFR3 mutant. (**B**) 293T cells were transfected with pKrox24 reporters together with empty vector and vectors expressing RTKs known to activate ERK pathway either weakly (INSR, IGF1R) or strongly (*FGFR1*, TRKB). Note the differences in pKrox24 transactivation (upper graphs), and EGR1 induction (lower blot), between strong and weak ERK activating RTKs. (Graphs) Values are averages from four biological replicates (each measured twice) with indicated S.D. Statistically significant differences are highlighted (Student's t-test; ***p<0.001). The levels of RTK expression were quantified by western blot with V5 antibody (lower blot). Actin serves as the loading control.**DOI:**
http://dx.doi.org/10.7554/eLife.21536.020

We equipped all the RTKs and their mutants cloned for this project with C-terminal V5 epitope for easy monitoring of expression of different RTKs via western blot with single commercial V5 antibody. Experiment comparing the EGR1 induction and Krox24 transactivation by unrelated RTKs belonging to different families is shown by the Figure 6 above. We demonstrate that equally expressed (as determined by V5 western blot) FGFR1, TRKB, INSR and IGF1R induce different levels of EGR1 expression and Krox24 transactivation, corresponding to their known ability to activate ERK pathway. Using the tools developed here, it is therefore possible to directly compare to what levels the different RTKs activate the common pathways of downstream signal transduction.

*3) FGFR inhibitors only partially block reporter activity caused by FGFR. Is this because of autocrine growth factor production and activation of ERK by another pathway? How does this limit the usefulness of the reporter?*

Manuscript Figure 1 demonstrates rather complete inhibition of Krox24 transactivation, mediated by FGFR3 transfection, with the cell treatment of FGFR inhibitor AZD1480. This is in contrast to data presented by Figure 1—figure supplement 5, which show only partial AZD1480-mediated inhibition of Krox24 transactivation caused by high endogenous FGFR activity in hESC and hiPSC cells. The data presented elsewhere (Gudernova et al., 2016; Human Molecular Genetics, 25, 9-23), or those on Figure 7 demonstrate that AZD1480 is a potent FGFR inhibitor, capable of complete suppression of activation of even highly active *FGFR1*-3 mutants. Thus the residual Krox24 activity after AZD1480 treatment, as shown in Figure 1—figure supplement 5, appears to be caused by ERK activation independent of FGFR signaling. This is no surprise since cultured cells usually have some levels of background ERK activity caused by serum growth factors. This fact could limit the use of Krox24 reporters in situations where the RTK transfection or activation of endogenous RTK signaling induces very little ERK activation on the top of high ERK background, and thus the two sources cannot be clearly distinguished from each other. Our data demonstrate that this is not the case for majority of the RTKs and their mutants tested here, which clearly induce EGR1 or transactivate Krox24 reporters when expressed in cells or endogenously, when activated by their cognate ligands (Figure 1,Figure 4,Figure 5; Figure 1—figure supplement 1, Figure 1—figure supplement 4, Figure 1—figure supplement 5, Figure 1—figure supplement 8; Figure 4—figure supplement 1; Figure 7).

Author response image 2.293T cells were transfected with empty vector or vector carrying the given FGFR variant, and treated with indicated AZD1480 for 24 hours.FGFR activation was determined by detection of FGFR auto-phosphorylation (**p**) at Tyr653/654. Actin serves as the loading control. Control, untransfected cells. Note the efficient inhibition of FGFR activation with AZD1480, even in the case of highly active FGFR2 and FGFR3 mutants N550K and K650M, respectively.**DOI:**
http://dx.doi.org/10.7554/eLife.21536.021

In the manuscript, the reporter transactivation data presented by Figure 1 and Figure 1—figure supplement 4, Figure 1—figure supplement 5, Figure 1—figure supplement 7 and Figure 1—figure supplement 8 were generated by treatment of three different cell types with fibroblast growth factor (FGF)-receptor ligand FGF2, and thus reflect the Krox24 reporter transactivation by endogenous FGFR signaling. In the revised manuscript, we evaluated the capacity of endogenous signaling of two other RTK families, EGFR and TRK, to transactivate Krox24 reporters. Figure 8 shows induction of EGR1 expression and transactivation of Krox24 reporters, mediated by endogenous EGFR signaling in 293T cells (activated by addition of EGF) or by endogenous TRK signaling in PC12 cells (activated by additions of NGF). Altogether, this demonstrates the capacity of Krox24 reporters to monitor the ligand-dependent activation of endogenous RTK signaling in cells.

Author response image 3.(**A**) 293T cells and (**B**) PC12 cells were transfected with pKrox24 reporters alone (293T) or together (PC12) with vector carrying *Renilla* luciferase under constant promoter (pRL-TK). Cells were treated with EGF (50 ng/ml) or NGF (100 ng/ml) for 24 hours before the level of Krox24 transactivation was determined by luciferase (**A**) or dual-luciferase (**B**) assay. Values are averages from four biological replicates (each measured twice) with indicated S.D. Statistically significant differences are highlighted (Student's t-test; ***p<0.001). Results are representative for three independent experiments. The levels of EGR1 induction by EGF or NGF treatment were determined by western blot. Actin serves as the loading control.**DOI:**
http://dx.doi.org/10.7554/eLife.21536.022

*2) To what extent do different kinases turn on EGR1? Is it possible to normalize for extent of expression and specific activity of the overexpressed kinase?*

*4) The use of over expression in 293T cells to provide ligand-independent signaling is potentially a problem because the constitutive activity of these ligand-regulated receptors may not reflect their physiological or pathophysiological activity? Thus, while the reporter assay may be useful for identifying inhibitors, one wonders if it may be limited in other regards. This issue deserves some comment.*

*5) The authors point to the uncertainty surrounding the clinical toxicity of the BCR-ABL inhibitor ponatinib (subsection “pKrox24 reporters can be used to identify novel targets for clinically used kinase inhibitors”, first paragraph) as a potential use for their assay, but do not indicate whether their analysis of ponatinib reveals new targets that might explain the side effects arising from its ability to target other RTKs.*

All clinically used BCR-ABL TKIs display some toxicity, which in the case of ponatinib resulted in recent discontinuation of the therapy. The analysis presented by manuscript Figure 5 reveals off-target activities for all five tested BCR-ABL TKIs, in agreement with the published literature ([Supplementary-material SD1-data]). Compared to other TKIs, ponatinib inhibits the largest amount of RTKs, which may account for its side effects in the chronic myeloid leukemia therapy, which exceed those of other TKIs. We however did not discover any novel ponatinib targets, all RTKs found inhibited here were already reported in the literature ([Supplementary-material SD1-data]). Instead, the experiments presented by Figure 5 and Figure 5—figure supplement 1 identified two new targets (LTK and INSR) for nilotinib. The revised manuscript text was updated to clearly indicate that, out of the five TKIs tested, novel targets were only identified for nilotinib (subsection “pKrox24 reporters can be used to identify novel targets for clinically used kinase inhibitors”, first paragraph).

We have chosen the BCR-ABL TKIs to demonstrate one possible application of the tools developed here because their off-target activities have been thoroughly evaluated before, and thus the published evidence may serve as a reference for estimation of the fidelity of developed screening assays. Our data demonstrate that the off-target activities of all five BCR-ABL TKIs found here match those previously reported, and that novel targets can be found even for the well-studied TKIs.

6) While the authors identify new targets for nilotinib (LTK and INSR), this discovery was made without the EGR1-based reporter. Inclusion of data for an application that utilized the EGR1-based reporter would be an effective way to demonstrate the utility of this system.

As suggested by published literature ([Supplementary-material SD1-data]), nilotinib and other four BCR-ABL TKIs evaluated in the study (Figure 5; Figure 5—figure supplement 1) inhibit many RTKs and thus any potential novel targets found in reporter screening would have to be validated and compared with known targets by independent and more precise experimental approach. To preserve time and resources, we therefore opted to skip the initial reporter screening and evaluated the TKI activity in experiments which allow determination of TKI effect on given RTK activity directly and in different TKI concentrations, as presented in Figure 5 and Figure 5—figure supplement 1.

The experiments presented by Figure 5 utilize the EGR1-based reporter to discover LTK as a novel target of EGFR inhibitor osimertinib, demonstrating the utility of Krox24 reporters for finding new targets for RTK inhibitors already used in clinic.